# The Impact of Point-of-Care Ultrasound on the Diagnosis and Management of Small Bowel Obstruction in the Emergency Department: A Retrospective Observational Single-Center Study

**DOI:** 10.3390/medicina60122006

**Published:** 2024-12-04

**Authors:** Carmine Cristiano Di Gioia, Alice Alame, Daniele Orso

**Affiliations:** 1Department of Emergency Medicine, Community Hospital of Baggiovara (MO), Azienda Ospedaliero-Universitaria di Modena, 41125 Modena, Italy; digioia.cristiano@aou.mo.it; 2Faculty of Medicine and Surgery, Department of Biomedical, Metabolic and Neural Sciences, University of Modena and Reggio Emilia, 41125 Modena, Italy; alice.alame@gmail.com; 3Department of Emergency, “Santa Maria della Misericordia” University Hospital of Udine, Azienda Sanitaria Universitaria Friuli Centrale, 33100 Udine, Italy

**Keywords:** small bowel obstruction, POCUS, emergency department, diagnosis, management

## Abstract

*Background and Objectives*: Small bowel obstruction (SBO) requires prompt diagnosis and management. Due to its advantages, POCUS can be beneficial when assessing SBO. However, it is still doubtful whether POCUS performed by an emergency doctor can prolong the time of patients with SBO in the emergency department (ED). The primary outcome was time to diagnosis when using POCUS compared to not using it. Secondary outcomes included the processing time in the ED, ED length of stay (LOS), rates of abdominal radiography, hospital LOS, and mortality. *Materials and Methods*: We conducted a retrospective, observational study in our ED from 1 November 2021 to 31 December 2023, including patients aged 18 and older diagnosed with SBO. Both groups received confirmation of their diagnosis through contrast-enhanced computed tomography. The two groups of patients (POCUS group vs. non-POCUS group) were compared regarding the time needed to reach the final diagnosis (i.e., time to diagnosis), the ED LOS, the hospital LOS, and in-hospital mortality. *Results*: A total of 106 patients were included. The median time to diagnosis was 121 min for the POCUS group vs. 217 min for the non-POCUS group (*p* < 0.001). Median ED processing time was 276 min in the POCUS group compared to 376 min in the non-POCUS group (*p* = 0.006). ED LOS was also shorter in the POCUS group (333 vs. 436 min, *p* = 0.010). Abdominal X-ray rates were lower in the POCUS group (49% vs. 78%, *p* = 0.004). Hospital LOS was similar between the two groups (*p* = 1.000). Five non-POCUS patients died during hospitalization; none died in the POCUS group, but the difference was not statistically significant (*p* = 0.063). *Conclusions*: POCUS significantly reduced time to diagnosis and ED LOS. Further exploration is needed to assess long-term outcomes and the cost-effectiveness of integrating POCUS into ED practice.

## 1. Introduction

Small bowel obstruction (SBO) is a common cause of emergency department (ED) visits and must be diagnosed and treated quickly due to its potential to be life-threatening [1]. The incidence in the USA is estimated to be between 2% and 4% in patients presenting to the ED for abdominal pain, and it is estimated to be 16% in surgical admissions [2]. The importance of timely diagnosis and management lies in reducing morbidity and mortality, preventing complications such as bowel strangulation, perforation, and peritonitis. Clinical evaluation and imaging tests, such as plain abdominal radiography (PAXR) and computed tomography (CT) scans [3], are currently used to diagnose SBO. In the ED, point-of-care ultrasound (POCUS) is a valuable diagnostic tool that allows for a rapid, non-invasive, and radiation-free evaluation of various abdominal pathologies, including SBO [4,5]. According to a recent expert consensus statement by Oto et al., the performance of POCUS should adhere to well-defined criteria, which include an adequate definition of the examination procedure and a clear distinction from a conventional ultrasound performed by a radiologist. In this context, the patient must be informed and educated about the functions and limitations of this methodology, and the performing physician must be adequately trained and certified [6].

POCUS performed by emergency physicians (EPs) has been demonstrated to be accurate in numerous studies [7,8], although its accuracy is dependent on the level of training and experience of the physician. Indeed, a recent meta-analysis found that POCUS exhibits high sensitivity (92%) and specificity (93%) in diagnosing SBO compared to the final clinical diagnosis [9]. 

The accuracy of POCUS in detecting SBO is comparable to that of ultrasound performed by a radiologist [10]. Furthermore, it has been suggested that POCUS is more cost-effective, leading to a decrease in overall healthcare costs compared to solely relying on CT imaging. Brower et al. report that using POCUS to diagnose SBO can result in savings of approximately USD 30.1 million annually by avoiding 143,000 CT scans [11]. While standard care approaches are still prevalent in the ED, POCUS has yet to be fully utilized as a clinical tool at the bedside. The current evidence on how POCUS can enhance the diagnosis and management of SBO in the ED is sparse. Existing studies focus on diagnostic accuracy measures such as sensitivity and specificity, without considering the impact on the overall patient care process. Their assumption is that there are potential benefits in the diagnostic workflow, but they fail to acknowledge that POCUS could potentially delay diagnosis instead of saving time and resources. The aim of this study is to assess the impact of POCUS on the diagnosis and management of SBO in the ED. The objective is to evaluate whether POCUS can assist emergency physicians in obtaining a more timely and appropriate diagnostic diagnosis, as well as enabling a more efficient use of available resources, compared to standard care alone.

## 2. Materials and Methods

### 2.1. Study Design

We conducted a retrospective, observational, single-center study enrolling consecutive patients with abdominal pain and clinical suspicion of SBO referred to the ED of Baggiovara Hospital in Modena, from 1st November 2021 to 31 December 2023. Individuals aged 18 years or older who were diagnosed with SBO using contrast-enhanced abdominal CT scan were eligible for inclusion. 

To minimize confounding factors and create more homogeneous patient groups for analysis, the following exclusion criteria were applied: pregnancy, palliative care patients, cases where obstruction was not the primary pathology in the context of multiple comorbidities and where surgery was not the principal treatment strategy, alternative diagnoses such as fecal impaction managed conservatively in the ED, and instances involving non-expert physician-performed evaluations.

Additional exclusion criteria included a prior diagnosis of SBO before ED presentation, transfer from another hospital for a CT scan, individuals with repeated SBO episodes in the past five years, and surgical anastomotic stenosis in patients awaiting corrective surgery, as these factors could lead to a more rapid diagnostic process. Delays associated with CT scanning procedures, such as shifts in radiologist schedules from nighttime to morning or technical equipment malfunctions, were also excluded since these factors would not be dependent on the emergency physician.

The patient population was divided into two groups for comparison reasons: the POCUS group and the non-POCUS group. The non-POCUS group typically performed conventional diagnostic workups, which usually involved PAXR and CT imaging.

### 2.2. POCUS Protocol

All participants followed a standardized protocol that included initial vital sign assessments at triage and clinical evaluations. Each participant was assigned a National Early Warning Score (NEWS) to categorize the clinical risk of patients.

The POCUS examinations were conducted by ED physicians who had completed a training and certification program in abdominal POCUS and had at least 5 years of extensive experience in this area, ensuring a consistent evaluation of the relevant parameters. The POCUS examinations of the intestinal loops were conducted using the ESAOTE MYLAB XPRO30 system (Esaote S.p.A., V. Siffredi, 58; 16,010 Genova, Italy), which was equipped with a 3.5 MHz convex transducer and a 7–12 MHz linear probe for detailed examination, as well as the portable Butterfly iQ ultrasound platform (Butterfly Network, Inc.; 1600 District Ave, Burlington, MA, USA). 

Using the standardized POCUS protocol, providers systematically examined the intestinal loops, starting from the right lower quadrant and following the path of the colon. The starting point was on the terminal ileum in the right lower quadrant. Then, we moved on to the cecum, ascending colon, transverse colon, descending colon, sigmoid colon, and finally the rectum. Subsequently, the small intestine in the upper quadrants and flanks was examined, using longitudinal and transverse imaging planes, in accordance with the American Gastroenterological Association guidelines and expert recommendations on intestinal ultrasound [12]. The diagnosis of SBO through POCUS relied on identifying at least three out of the four ultrasound indicators that are commonly linked to the highest specificity and sensitivity in diagnosing SBO via POCUS: (1) intestinal loops > 2.5 cm in diameter; (2) presence of free fluid between the loops (known as “tanga sign”); (3) observed alternating peristalsis (“to-and-fro” movement); and (4) visualization of circular folds extending perpendicular to the wall of the duodenum and jejunum (referred to as “keyboard sign”) [4] (Figure 1).

### 2.3. Outcomes

The primary endpoint of this study was the estimation of the difference in the time to diagnosis of SBO, defined as the time from the start of the emergency physician’s assessment to the time that the diagnosis of SBO was confirmed by CT scan (the execution time of CT scan). The secondary outcome included the estimation of differences in the ED processing time, in the ED LOS, in the abdominal radiography rates, in the hospital LOS, and in mortality. The ED processing time ranges from the initial physical examination to surgery or hospital admission. The ED LOS was the time that patients spent in the ED from their arrival in triage until discharge. Hospital LOS was defined as the period between admission and discharge from the hospital. Mortality was defined as the occurrence of death during the patient’s hospitalization (Figure 2). 

### 2.4. Data Management

The electronic medical records of each patient were analyzed to obtain the data. A standardized form was used by a student and an emergency medicine resident to extract relevant demographic and clinical information after a two-hour training session, and this information included vital parameters upon arrival in the ED, clinical history, an objective examination conducted by the on-duty physician, instrumental examinations, time to assess SBO presence, ED processing time, ED LOS, hospital LOS, and mortality. 

### 2.5. Statistical Analysis

Descriptive statistics for all variables were calculated using Microsoft Excel (Microsoft Corp, One Microsoft Way, Redmond, WA, USA) and R version 4.3.2 (R Core Team (2023). R: A Language and Environment for Statistical Computing. R Foundation for Statistical Computing, Vienna, Austria). The variables expressed as frequencies have been transformed into percentages or ratios. To verify the normal distribution of data, a Shapiro–Wilk test was carried out for every comparison between the two groups. A Mann–Whitney test was used to compare the two groups because the distribution was non-parametric.

The degree of dependence on the ED processing time and the time to diagnosis was determined through a linear regression test by subdividing the two study groups (POCUS vs. non-POCUS).

Emmeans test was employed to carry out a pairwise comparison of multiple groups using Bonferroni multiplicity correction to verify the distribution of NEWS grade in relation to the two study groups and the time to diagnosis. An EMMs test (i.e., emmeans test), commonly referred to as least-squares means in traditional regression models, is calculated by utilizing a model to predict over a regular grid of predictor combinations (i.e., time to diagnosis and NEWS).

For all comparisons, an alpha error of 5% (*p*-value = 0.05) was considered the threshold for statistical significance.

## 3. Results

A total of 325 patients had accessed the ED for a diagnosis of an abdominal emergency equivalent. Of these, 219 were excluded, leaving 106 patients for analysis (Figure 3). The median age of patients was 76 years (IQR 67.2–82.0), and 50.5% were females. Of the 106 patients included in the final analysis, 47 (44%) were subjected to POCUS; 59 (56%) were not.

The median time to diagnosis was 217 min (95%CI 188–246) in the non-POCUS group and 121 min (95%CI 107–135) in the POCUS group (*p* < 0.001) (Figure 4). 

The median ED processing time was 376 min (95%CI 321–431) in the non-POCUS group and 276 min (95%CI 229–322) in the POCUS group (*p* = 0.006). The median overall ED LOS was 436 min (95%CI 377–494) in the non-POCUS group and 333 min (95%CI 281–385) in the POCUS group (*p* = 0.010) (Table 1). 

There was an imbalance in the distribution of NEWS classes between the two groups. The analysis of time to diagnosis compared to the NEWS classes showed a tendency to a reduction in time in the POCUS group as the NEWS class increased; however, the pairwise comparison was not statistically significant (*p* = 0.490) (Appendix A).

Of the 47 patients in the POCUS group, 23 (49%) had a PAXR before CT scan. Out of the 59 patients in the non-POCUS group, 46 (78%) underwent PAXR prior to CT scan (*p*-value = 0.004) (Table 1). 

The two study groups had different correlations between time to diagnosis and processing time. In the non-POCUS group, the correlation was r = 0.613, while in the POCUS group, it was r = 0.333 (Figure 5).

There was no significant difference in the hospital LOS between the two groups (*p* = 1.000). 

Five patients from the non-POCUS group died during hospitalization, whereas no patients from the POCUS group died (Figure 6). However, the difference was not statistically significant (*p* = 0.063).

## 4. Discussion

This study demonstrates the significance of POCUS evaluation of intestinal loops for patients with suspected SBO. POCUS has been employed by EPs in the assessment of patients with suspected SBO as a rapid and economical diagnostic tool that can be carried out at the bedside without the use of radiation. POCUS has been extensively employed in research to detect SBO, with a sensitivity and specificity that is equivalent to that of CT scans [9,13,14]. However, the specificity of POCUS is significantly lower when performed in the ED setting and when CT scans are used as the sole reference standard for confirming the diagnosis [9,15]. Blanco and Volpicelli emphasized that POCUS is a valuable tool, but it requires adequate training and consistent practice for proper interpretation to avoid misdiagnosis [16]. Schott et al. identified significant barriers to POCUS implementation in ICUs, including training deficiencies (i.e., lack of trained personnel, funding, and opportunities), infrastructural limitations (i.e., image archiving, championing, and resources), and equipment shortages. To maximize POCUS utilization and improve patient care, it is crucial to address these obstacles. These barriers hinder widespread adoption of POCUS, despite its recognized clinical value [17]. Although there are many advantages mentioned in the literature, POCUS has not been widely implemented as a standard practice in the ED. The diagnostic approach for SBO was enhanced by EP with extensive POCUS expertise, according to our study. POCUS enabled a faster diagnosis of SBO compared to the traditional approach of using plain abdominal radiographs and CT scans.

SBO is associated with high morbidity and mortality, and a delay in the diagnosis is linked to higher mortality due to complications such as bowel necrosis, perforation, sepsis, and septic shock [18]. Singer et al. demonstrated that POCUS reduced the time to diagnose SBO compared to CT with a mean time of 130 min vs. 297 min [15]. Performing a POCUS before a CT scan resulted in a quicker diagnosis (121 min compared to 217 min), and our study confirmed the effectiveness of ultrasound as a preliminary diagnostic tool. The rapid planning and execution of a CT scan following a suspicion of SBO on POCUS may have contributed to the observed results. Furthermore, a quicker consultation with a radiologist and surgical referral could speed up the diagnostic and management process. Our findings indicate that the POCUS group had a shorter ED LOS compared to the non-POCUS group (333 min vs. 436 min). This results in a faster surgical consultation and admission process, which reduces the workload for ED staff.

Guttman and colleagues reported a case in which the early use of ultrasound led to a swift diagnosis of SBO, allowing for the timely transfer to the surgical department for proper management [19]. Our findings indicate that an ultrasound-based approach could have a significant impact on patient flow in the ED, despite Guttman not providing data on management times.

Lindelius et al. found no significant difference in ED LOS between the two groups [20]. The ED’s organizational differences may have contributed to this finding. The current literature does not cover the impact of POCUS on ED LOS. Brower et al. developed a simulation that utilized a model to estimate the potential decrease in ED LOS by conducting bowel loop POCUS prior to a CT scan in patients with SBO [11]. The simulation led to a total decrease of 507,000 h in ED LOS. Despite what we found, a recent study found that POCUS does not have a significant impact on the ED LOS in patients with non-traumatic acute abdominal pain [21]. The physicians who perform the ultrasound and their focus on SBO could play a significant role. We observed that in the POCUS group, approximately one-third of the processing time was used to reach the diagnosis, while the remaining two-thirds were allocated to admitting the patients to the hospital. The correlation between time to diagnosis and ED processing time showed that the non-POCUS group took almost twice as long to reach diagnosis. Furthermore, after performing POCUS, medical staff typically request a CT scan directly without first ordering an X-ray. This hypothesis is supported by the results, with only 49% of patients in the POCUS group receiving an X-ray compared to 78% of patients in the non-POCUS group. The positive linear correlation between processing time and time to diagnosis can also be explained by these results.

The request for extra diagnostic imaging tests, such as X-ray and CT scans, did not differ significantly in the study by Lindelius et al. [20]. According to our study, there has been a significant decrease in the number of requested X-rays in the POCUS group (*p* = 0.004). This difference may be since the study by Lindelius et al. also included patients with acute abdominal pain who did not have SBO and therefore required different types of diagnostic tests to characterize their condition. The study by Lindelius et al. involved surgeons evaluating patients after completing ultrasound training for three weeks and only four weeks working in the ED. In our study, the ultrasounds were performed by experienced EPs in POCUS in emergency settings. It can be hypothesized that in our ED’s clinical practice, when SBO is suspected, POCUS is increasingly being preferred over PAXR whenever possible. This aligns with the existing literature that demonstrates the diagnostic superiority of ultrasound. According to Jang and colleagues, PAXR was able to diagnose SBO with a sensitivity and specificity of 46% and 67%, compared to 94% and 81% for POCUS [22]. Moreover, abdominal radiography was non-diagnostic in 36% of the cases, while none of the ultrasounds were evaluated as indeterminate [22]. Taylor and Lalani suggested that PAXR was the least suitable for diagnosis, with a positive likelihood ratio (+LR) of 1.64. Bedside POCUS was found to be a reliable approach, with a +LR of 9.55 and a negative likelihood ratio of 0.04 for ED residents [23].

Our study, in line with other research groups, confirms that there was no significant difference in hospital LOS [20]. The hospital LOS of patients with acute abdominal pain and complications related to the pathological condition cannot be significantly influenced by the imaging techniques used but more by the time spent in management in hospital wards or surgery theater.

Five patients died in the non-POCUS group due to complications caused by their pathological condition, while there were no deaths in the POCUS group. This finding confirms the safety of POCUS, as described by Lindelius and colleagues [20], and shows that it does not harm the prognosis of patients evaluated in the ED. POCUS is at risk for potential misdiagnoses because of false negatives and false positives [24]. Experienced personnel can use a three-step approach (clinical examination, POCUS, and CT scan) to diagnose patients quickly without increasing patient mortality. 

Our study demonstrates the significant advantages of including POCUS in the clinical management of patients with suspected SBO. 

When performed by experienced EPs trained in this technique, POCUS can significantly reduce the time these patients spend in the ED and improve their diagnosis accuracy and timeliness.

By providing more expedient and direct diagnostic support, POCUS enables earlier recognition of SBO, which may lead to improved patient outcomes. In addition, this approach assists in decreasing the duration of stay for SBO patients in the ED, which alleviates the burden on healthcare providers and reduces the risks associated with extended waiting times.

### Limitations

Our study is subject to several limitations. The retrospective nature of this study could have led to incomplete data. Furthermore, this study depended on chart review and data retrieval, which could have been subject to bias. We tried to obtain as much data as possible from the study patients. However, we failed to assess the EP’s reason for choosing one imaging technique over another.

On the other hand, the retrospective study design allowed us to eliminate the Hawthorne effect and to analyze what happens in clinical practice in the ED. Nonetheless, a prospective study that considers confounding factors and tries to eliminate them may be necessary. 

The data could have been impacted by the physicians’ level of experience, leading to a new confounding factor. All examinations were carried out by certified EPs who completed training during their residency.

This study’s single-center location may prevent the data from being generalizable at a national level. In fact, the level of experience and competence in the use of ultrasound is not homogenous among emergency departments at national level. To verify the reproducibility of our experience, a multicenter prospective study would be necessary.

## 5. Conclusions

The time to diagnose SBO in patients with non-traumatic abdominal pain presented to the ED was reduced by using POCUS. While CT scans are still necessary to fully characterize the cause, POCUS that is focused on bowel obstruction facilitates the decision-making and patient management process, which reduces the length of stay in the ED. By identifying characteristic signs of SBO through ultrasound, the need for a pre-CT X-ray can be avoided, which can reduce radiation exposure, diagnostic time, and cost. In addition, POCUS is a technique that is safe, rapid, and repeatable and does not have any negative impact on clinical outcomes. According to these findings, POCUS should be used during the initial assessment of patients who come to the ED with acute abdominal pain. To validate these results, additional prospective, multicenter investigations would be needed. Such studies should aim to further evaluate the use of POCUS in the diagnosis and management of SBO, analyzing its impact on clinical outcomes, resource utilization, and patient flow in the ED. Prospective studies with larger sample sizes that account for potential confounding factors would provide more robust and generalizable evidence to guide clinical practice.

## Figures and Tables

**Figure 1 medicina-60-02006-f001:**
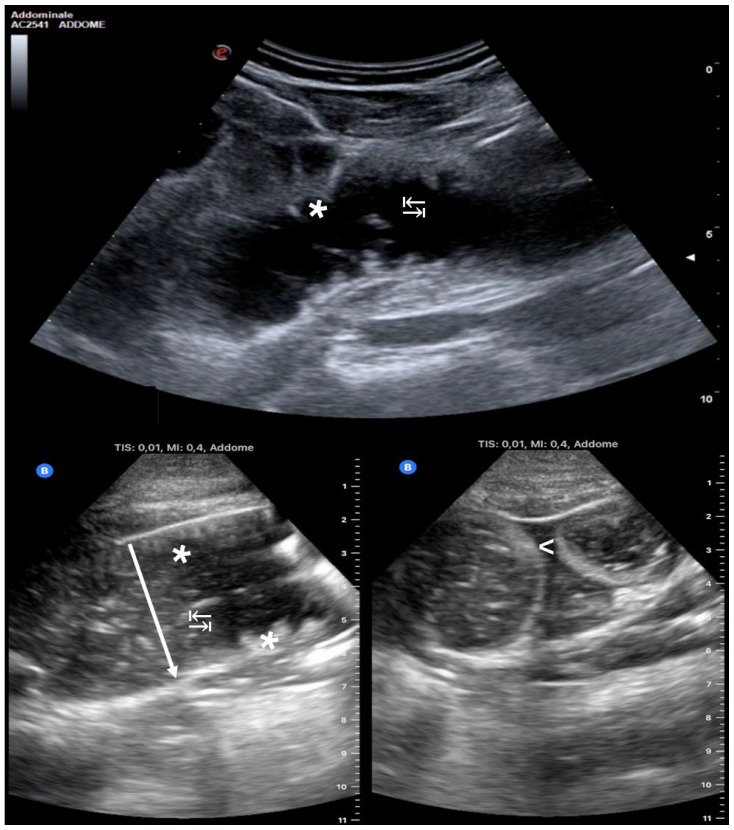
The ultrasound images depict four key diagnostic indicators of SBO. Distended intestinal loops (↓) in the figure lower left: with a diameter exceeding the physiological threshold for distension, set at 2.5 cm; the “tanga sign” (<): presence of fluid between the bowel loops with a characteristic triangular “thong-shaped” distribution; the “keyboard sign” (*): hypertrophy of Kerckring folds, typically visible in the jejunum (more numerous) and ileum (less numerous) due to the pathophysiology of SBO; and the “to-and-fro sign” (↹): alternating peristaltic movement caused by downstream obstruction.

**Figure 2 medicina-60-02006-f002:**
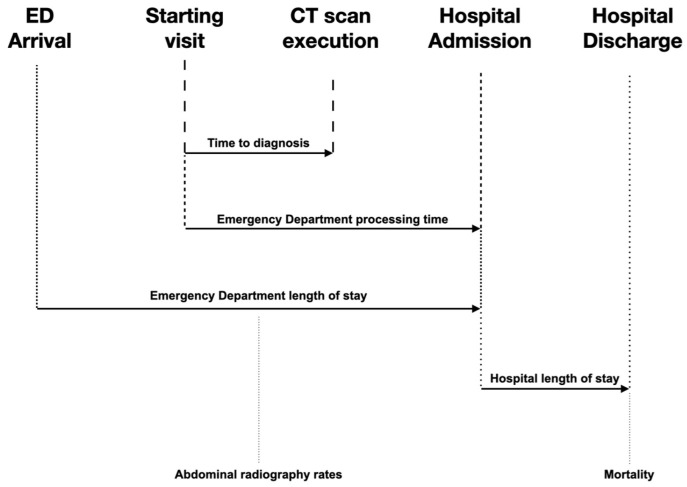
Primary and secondary outcomes. Time to diagnosis (from the initial visit to the completion of CT scan), processing time in the emergency department (from the initial visit to hospital admission), emergency department length of stay (from triage to discharge from the ER), hospital length of stay (from hospital admission to discharge), abdominal radiography rates, and mortality (during the patient’s hospital stay).

**Figure 3 medicina-60-02006-f003:**
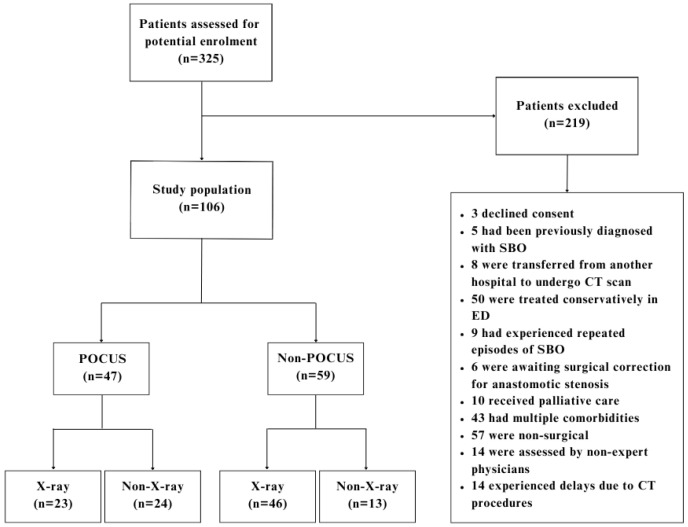
Flowchart illustrating the selection of the study samples. POCUS, point-of-care ultrasound.

**Figure 4 medicina-60-02006-f004:**
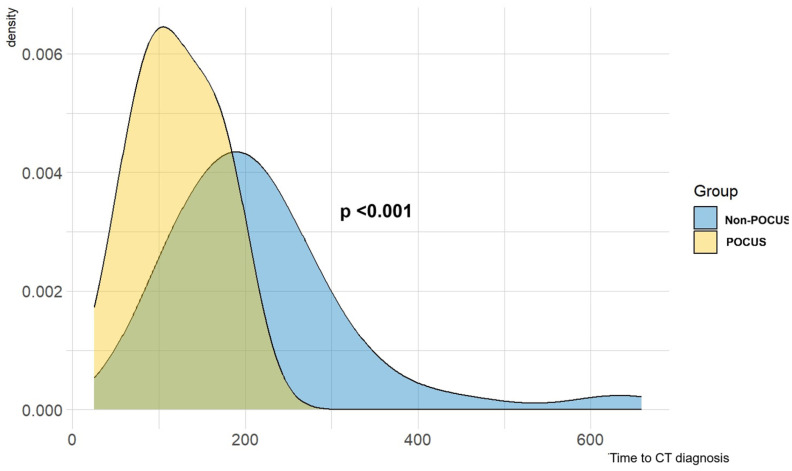
Difference in times for CT diagnosis between the two groups. The diagnosis was reached more quickly by the POCUS group (121 vs. 217 min; *p* < 0.001).

**Figure 5 medicina-60-02006-f005:**
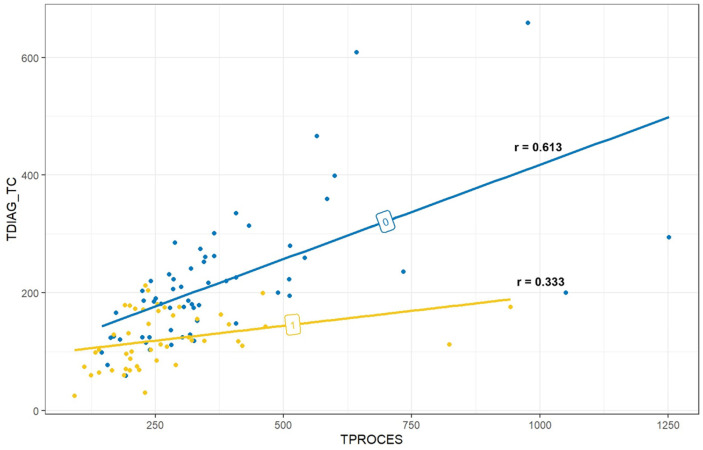
Correlation between the time to diagnosis (on axis y) and the ED processing time (on axis x). The correlation is stronger in the non-POCUS group (r = 0.61 vs. 0.33). The time taken to arrive at diagnosis in the POCUS group has less impact on overall processing time than in the non-POCUS group. Group 1: POCUS group; Group 0: non-POCUS group.

**Figure 6 medicina-60-02006-f006:**
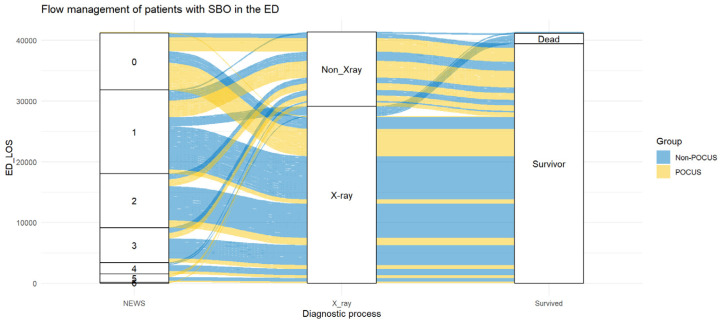
The management flow of patients included in this study. In the non-POCUS group, 5 deaths occurred; in the POCUS group, no deaths occurred.

**Table 1 medicina-60-02006-t001:** Distribution of the main outcome variables between the two groups.

		POCUS Group	Non-POCUS Group	*p*-Overall
		N = 47	N = 59	
NEWS				0.015
	0	16 (34.8%)	6 (10.2%)	
	1	11 (23.9%)	25 (42.4%)	
	2	8 (17.4%)	13 (22.0%)	
	3	6 (13.0%)	10 (16.9%)	
	4	1 (2.2%)	4 (6.8%)	
	5	3 (6.5%)	1 (1.7%)	
	6	1 (2.2%)	0 (0.00%)	
PAXR		23 (48.9%)	46 (78.0%)	0.004
Time to CT diagnosis (minutes)		121 [107–135]	217 [188–246]	<0.001
ED processing time (minutes)		276 [229–322]	376 [321–431]	0.006
ED LOS (minutes)		333 [281–385]	436 [377–494]	0.01
Hospital LOS (days)		12.3 [9.15–15.5]	12.3 [8.86–15.8]	1
Deceased		0 (0.00%)	5 (8.62%)	0.063

Continuous variables are expressed as mean ± standard deviation. In round brackets, the relative percentages; in square brackets, the 95% confidence interval. POCUS: point-of-care ultrasound; NEWS: National Early Warning Score; PAXR: plain abdominal X-ray; CT: computed tomography; ED: emergency department; LOS: length of stay.

## Data Availability

Data may be provided by the authors for reasonable reasons.

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
