# Peer review of "The Impact of Point-of-Care Ultrasound on the Diagnosis and Management of Small Bowel Obstruction in the Emergency Department: A Retrospective Observational Single-Center Study"

_medicina, 2024, doi:10.3390/medicina60122006_

Round 1

Reviewer 1 Report

Comments and Suggestions for Authors

Abstract

Lines 13-31: The abstract is concise, but it could be improved by specifying the statistical tests used to validate the results. This will give the reader a better idea of the rigor of the analysis. Consider mentioning this briefly.

Introduction

Lines 35-62: The introduction provides a strong background, but the transition from the general importance of SBO to the study objectives could be smoother. Consider connecting these ideas more clearly, perhaps by summarizing why previous studies have been insufficient.

Methods

1. Study Design:

Lines 63-79: The description of the study design is comprehensive but could benefit from more clarity. For instance, it might be helpful to explain why the exclusion criteria were chosen. Consider adding a justification for each criterion to reinforce the study’s robustness.

2. POCUS Protocol:

Lines 85-109: The protocol is well-documented. However, it would be useful to mention how the training of physicians is standardized. Were there any measures to ensure inter-rater reliability? This would strengthen the credibility of the methodology.

3. Statistical Analysis:

Lines 142-151: While the analysis methods are sound, the description of statistical tests is brief. Specify why you chose the Shapiro-Wilk test for normality and clarify the rationale for using Tukey testing with Bonferroni correction. This might help readers understand the approach better.

Results

Lines 152-202: The results are clearly presented. However, the study could benefit from more visual aids. Consider adding an additional table or figure to summarize key statistical differences more effectively.

Table 1: The data in Table 1 is informative, but adding confidence intervals for the continuous variables could provide more depth. This would be especially useful for readers interested in the precision of your estimates.

Discussion

1. Interpretation of Findings:

Lines 203-250: The discussion is well-structured, but it relies heavily on comparisons to previous studies. It would be beneficial to elaborate more on the clinical implications of your findings and how they might influence practice in emergency departments.

2. Limitations:

Lines 267-276: The limitations section is strong. However, consider addressing the potential bias introduced by the retrospective design in more depth. Were there any strategies you used to mitigate this bias? Additionally, discussing the implications of not generalizing the results to a national level would be helpful.

Conclusion

Lines 277-287: The conclusion is clear, but it might be worth emphasizing the need for future research to validate these findings further. You could also mention any plans for a follow-up study if applicable.

General Suggestions:

1. Clarity: In some areas, the language is slightly technical and may be difficult for a broader audience to follow. Simplifying or explaining complex terms could make the paper more accessible.

2. Figures: The figures included are helpful, but you might want to add more annotations to guide the reader through the important features highlighted in the ultrasound images.

3. Consistency: Ensure consistency in reporting statistical values. For example, always report p-values with the same number of decimal places for uniformity.

Author Response

We are grateful to the editor and reviewers for their attention to our work and for the constructive comments they have made. We answered point by point.

Reviewer #1

Abstract

  • Lines 13-31: The abstract is concise, but it could be improved by specifying the statistical tests used to validate the results. This will give the reader a better idea of the rigor of the analysis. Consider mentioning this briefly.

We added the following sentence in the Abstract to better define the comparisons made between the two study groups:

“The two groups of patients (POCUS group vs non-POCUS group) were compared regarding the time needed to reach the final diagnosis (ie, time to diagnosis), the ED LOS, the hospital LOS and in-hospital mortality.”

Introduction

  • Lines 35-62: The introduction provides a strong background, but the transition from the general importance of SBO to the study objectives could be smoother. Consider connecting these ideas more clearly, perhaps by summarizing why previous studies have been insufficient.

We attempted to improve the flow of the text in the introduction by replacing the content  as follows:

While standard care approaches remain common in the ED, the effective use of POCUS as a bedside clinical tool has not been fully realized. Current evidence on how POCUS can improve the diagnosis and management of SBO in the ED is limited. Existing studies focus on diagnostic accuracy measures like sensitivity and specificity, without considering the impact on the overall patient care process. They only assume potential benefits in the diagnostic workflow, without overcoming the perception that POCUS could potentially delay diagnosis rather than save time and resources. This study aims to evaluate the influence of POCUS on the diagnosis and management of SBO in the ED. It will assess whether POCUS can help emergency physicians reach a more prompt and appropriate diagnostic approach, as well as enable more efficient use of available resources, compared to standard care alone.

Methods

  1. Study Design:
  • Lines 63-79: The description of the study design is comprehensive but could benefit from more clarity. For instance, it might be helpful to explain why the exclusion criteria were chosen. Consider adding a justification for each criterion to reinforce the study’s robustness.

We attempted to better outline and add the individual reasons regarding the exclusion criteria.

To minimize confounding factors and create more homogeneous patient groups for analysis, the following exclusion criteria were applied: pregnancy, palliative care patients, cases where the obstruction was not the primary pathology in the context of multiple comorbidities and where surgery was not the principal treatment strategy, alternative diagnoses such as fecal impaction managed conservatively in the ED, and instances involving non-expert physician-performed evaluations.

Additional exclusion criteria included prior diagnosis of SBO before ED presentation, transfer from another hospital for a CT scan, individuals with repeated SBO episodes in the past five years, and surgical anastomotic stenosis in patients awaiting corrective surgery, as these factors could lead to a more rapid diagnostic process. Delays associated with CT scanning procedures, such as shifts in radiologist schedules from nighttime to morning or technical equipment malfunctions, were also excluded since these factors would not be dependent on the emergency physician.

  1. POCUS Protocol:
  • Lines 85-109: The protocol is well-documented. However, it would be useful to mention how the training of physicians is standardized. Were there any measures to ensure inter-rater reliability? This would strengthen the credibility of the methodology.

We provided a more detailed specification of the training criteria for physicians performing abdominal POCUS.

The POCUS examinations were conducted by ED physicians who had completed a training and certification program in abdominal POCUS and had at least 5 years of extensive experience in this area, ensuring a consistent evaluation of the relevant parameters.

  1. Statistical Analysis:
  • Lines 142-151: While the analysis methods are sound, the description of statistical tests is brief. Specify why you chose the Shapiro-Wilk test for normality and clarify the rationale for using Tukey testing with Bonferroni correction. This might help readers understand the approach better.

We have expanded the subsection on statistical methods. We also noticed some errors in the definition: we used averages and not medians for continuous variables. And the comparison between groups was made using an emmeans test. The subsection is thus:

“Descriptive statistics for all variables were calculated using Microsoft Excel (Microsoft Corp) and R version 4.3.2 (R Core Team (2023). R: A Language and Environment for Statistical Computing. R Foundation for Statistical Computing, Vienna, Austria). The variables expressed as frequencies have been transformed into percentages or ratios. To verify the normal distribution of data, a Shapiro-Wilk test was carried out for every comparison between the two groups. A Mann-Whitney test was used to compare the two groups, because the distribution was non-parametric.

 The degree of dependence on the ED processing time and the time-to-diagnosis was determined through a linear regression test by subdividing the two study groups (POCUS vs non-POCUS).

Emmeans test was employed to carry out a pairwise comparison of multiple groups using Bonferroni multiplicity correction to verify the distribution of NEWS grade in relation to the two study groups and the time to diagnosis. EMMs test (ie emmeans test), commonly referred to as least-squares means in traditional regression models, is calculated by utilizing a model to predict over a regular grid of predictor combinations (ie, time-to-diagnosis and NEWS).

For all comparisons, an alpha error of 5% (p-value = 0.05) was considered as the threshold for statistical significance.”

Results

  • Lines 152-202: The results are clearly presented. However, the study could benefit from more visual aids. Consider adding an additional table or figure to summarize key statistical differences more effectively.

We reported the difference in diagnosis time (primary outcome) using a density plot. The remaining secondary outcomes were represented by linear correlation plots or boxplots. We have not been able to get more explanatory graphs than these but are ready to correct according to the suggestions of the reviewers.

  • Table 1: The data in Table 1 is informative, but adding confidence intervals for the continuous variables could provide more depth. This would be especially useful for readers interested in the precision of your estimates.

We have been able to show the dispersion of continuous variables by confidence interval.

Discussion

  1. Interpretation of Findings:
  • Lines 203-250: The discussion is well-structured, but it relies heavily on comparisons to previous studies. It would be beneficial to elaborate more on the clinical implications of your findings and how they might influence practice in emergency departments.

In the last paragraph of the discussion, we added the following:

Our study demonstrates the substantial benefits of incorporating POCUS into the clinical management of patients with suspected SBO. POCUS has been found to significantly reduce the time spent by these patients in the ED, while also improving the accuracy and timeliness of their diagnosis. By providing more expedient and direct diagnostic support, POCUS enables earlier recognition of SBO, which may lead to improved patient outcomes. Furthermore, this approach helps to decrease the length of stay for SBO patients in the ED, thereby alleviating the burden on healthcare providers and mitigating the risks associated with extended waiting times.

  1. Limitations:
  • Lines 267-276: The limitations section is strong. However, consider addressing the potential bias introduced by the retrospective design in more depth. Were there any strategies you used to mitigate this bias? Additionally, discussing the implications of not generalizing the results to a national level would be helpful.

We have further specified some of the limitations of our study, trying to explain the corrective strategies accomplished. The subsection therefore reads as follows:

“Our study is subject to several limitations. The retrospective nature of the study could lead to incomplete data. Furthermore, the study depended on chart review and data retrieval, which could be subject to bias. We tried to obtain as much data as possible from the study patients. However, we failed to assess the EP's reason for choosing one imaging technique over another.

On the other hand, the retrospective study design allowed us to eliminate the Hawthorne effect and to analyze what happens in clinical practice in the ED. Nonetheless, a prospective study that considers confounding factors and tries to eliminate them may be necessary.

The data could be impacted by the physicians' level of experience, leading to a new confounding factor. All examinations were carried out by certified EPs who completed training during their residency.

The study's single center location may prevent the data from being generalizable at a national level. In fact, the level of experience and competence in the use of ultrasound is not homogenous among emergency departments at national level. To verify the reproducibility of our experience, a multicentre prospective study would be necessary.”

Conclusion

  • Lines 277-287: The conclusion is clear, but it might be worth emphasizing the need for future research to validate these findings further. You could also mention any plans for a follow-up study if applicable.

We added this additional specification in the conclusions:

To validate these results, additional prospective, multi-center investigations would be needed. Such studies should aim to further evaluate the use of POCUS in the diagnosis and management of SBO, analyzing its impact on clinical outcomes, resource utilization, and patient flow in the ED. Prospective studies with larger sample sizes that account for potential confounding factors would provide more robust and generalizable evidence to guide clinical practice.

General Suggestions:

  1. Clarity: In some areas, the language is slightly technical and may be difficult for a broader audience to follow. Simplifying or explaining complex terms could make the paper more accessible.

We have revised the entire manuscript linguistically, trying to make the description of our study clearer and more precise.

  1. Figures: The figures included are helpful, but you might want to add more annotations to guide the reader through the important features highlighted in the ultrasound images.

We added this description:

Distended intestinal loops (↓): with a diameter exceeding the physiological threshold for distension, set at 2.5 cm, the "tanga sign" (<): presence of fluid between the bowel loops with a characteristic triangular “thong-shaped” distribution, the "keyboard sign" (*): hypertrophy of the Kerckring folds, typically visible in the jejunum (more numerous) and ileum (less numerous) due to the pathophysiology of SBO, and the "to-and-fro sign" (↹): alternating peristaltic movement caused by downstream obstruction.

  1. Consistency: Ensure consistency in reporting statistical values. For example, always report p-values with the same number of decimal places for uniformity.

We have revised the entire manuscript linguistically, trying to make the description of our study clearer and more precise.

Reviewer 2 Report

Comments and Suggestions for Authors

This article presented the usefulness of point-of-care ultrasound (POCUS) in managing small bowel obstruction in the ED. It found that POCUS reduced the time to diagnosis and length of stay (LOS). However, there are concerns about this article. 1. The primary outcome of this study includes the time of diagnosis. However, it depends on the performer's skills and conditions. Therefore, in my view, the time of diagnosis could have belonged to secondary outcomes. 2. Could you explain what US skills are? Who performs US in the ED? 3. Talel1 needs an explanation of the abbreviation. 3. They determine the execution time of the CT scan. Please provide the approximate average duration of the CT scan.

Comments on the Quality of English Language

Minor English editing is requierd.

Author Response

(The authors gave the same response as above.)

Round 2

Reviewer 1 Report

Comments and Suggestions for Authors

It fits scientific expectations.